# Host autophagy machinery is diverted to the pathogen interface to mediate focal defense responses against the Irish potato famine pathogen

Yasin F Dagdas[1,2†], Pooja Pandey[3†], Yasin Tumtas[3], Nattapong Sanguankiattichai[3], Khaoula Belhaj[1], Cian Duggan[3], Alexandre Y Leary[3], Maria E Segretin[4], Mauricio P Contreras[3,4], Zachary Savage[3], Virendrasinh S Khandare[3], Sophien Kamoun[1]*, Tolga O Bozkurt[3]*

[1]The Sainsbury Laboratory, Norwich Research Park, Norwich, United Kingdom; [2]The Gregor Mendel Institute of Molecular Plant Biology, Vienna Biocenter, Vienna, Austria; [3]Department of Life Sciences, Imperial College London, London, United Kingdom; [4]INGEBI-CONICET, Ciudad Autonoma de Buenos Aires, Buenos Aires, Argentina

**\*For correspondence:**
sophien.kamoun@tsl.ac.uk (SK);
o.bozkurt@imperial.ac.uk (TOB)

[†]These authors contributed equally to this work

**Abstract** During plant cell invasion, the oomycete *Phytophthora infestans* remains enveloped by host-derived membranes whose functional properties are poorly understood. *P. infestans* secretes a myriad of effector proteins through these interfaces for plant colonization. Recently we showed that the effector protein PexRD54 reprograms host-selective autophagy by antagonising antimicrobial-autophagy receptor Joka2/NBR1 for ATG8CL binding (Dagdas et al., 2016). Here, we show that during infection, ATG8CL/Joka2 labelled defense-related autophagosomes are diverted toward the perimicrobial host membrane to restrict pathogen growth. PexRD54 also localizes to autophagosomes across the perimicrobial membrane, consistent with the view that the pathogen remodels host-microbe interface by co-opting the host autophagy machinery. Furthermore, we show that the host-pathogen interface is a hotspot for autophagosome biogenesis. Notably, overexpression of the early autophagosome biogenesis protein ATG9 enhances plant immunity. Our results implicate selective autophagy in polarized immune responses of plants and point to more complex functions for autophagy than the widely known degradative roles.
DOI: https://doi.org/10.7554/eLife.37476.001

## Introduction

Plants intimately interact with a diverse range of pathogens, which typically produce specialized structures, such as haustoria, to invade the host cell space (*Panstruga and Dodds, 2009*). These specialized structures are surrounded by membranes derived from the host endomembrane system (*Bozkurt et al., 2015, 2014*; *Whisson et al., 2016*; *Bozkurt et al., 2011*), which mediate inter-organismal communication enabling nutrient and macromolecule trafficking (*Wang et al., 2017*; *Dagdas et al., 2016*; *Micali et al., 2011*; *Koh et al., 2005*; *Le Fevre et al., 2015*; *Gutjahr and Parniske, 2013*; *Pumplin et al., 2012*). However, our understanding of the origin and biogenesis of these host-microbe interfaces remains limited. In particular, the extent to which host accommodation membranes are shaped by the invading microbes is unclear.

Similar to many other filamentous plant pathogens, the potato late blight pathogen *Phytophthora infestans* produces haustoria, hyphal extensions that invaginate the host cell membrane (*Whisson et al., 2016*). Strikingly, the host accommodation membrane, also known as the

extrahaustorial membrane (EHM) sharply contrasts with the adjacent plasma membrane in both protein and lipid composition (*Bozkurt et al., 2014*; *Lu et al., 2012*; *Schornack et al., 2009*). Thus, the EHM is not uniform, and multiple membrane sources probably contribute to its biogenesis (*Bozkurt et al., 2015*). The emerging model is that multiple trafficking pathways are diverted to haustoria with some degree of specificity.

Macroautophagy (hereafter called autophagy) is an evolutionary conserved membrane trafficking pathway that mediates removal or relocation of cytoplasmic components (*Stolz et al., 2014*). Bulk autophagy, typically activated by starvation, involves non-selective engulfment of cytoplasmic materials to double membrane vesicles called autophagosomes, which are then carried to the vacuole for recycling. In contrast, selective-autophagy employs specialized autophagy cargo receptors that bind ATG8 on autophagosome membranes, and recruit specific cargoes to autophagosomes (*Lamb et al., 2013*). For instance, to destroy viral particles and restrict viral infection, the plant autophagy cargo receptor Joka2/NBR1 activates antimicrobial autophagy, also known as xenophagy (*Hafrén et al., 2017*). Similarly, we recently showed that Joka2/NBR1 mediated selective autophagy pathway contributes to defense against *P. infestans.* However, the molecular basis of this Joka2/NBR1 mediated defense-related autophagy remains unknown (*Dagdas et al., 2016*). To counteract selective autophagy, *P. infestans* deploys PexRD54, a secreted protein that belongs to the large RXLR-WY family of virulence effectors (*Dagdas et al., 2016*). PexRD54 carries an ATG8 interacting motif (AIM) and attenuates defense-related autophagy by depleting Joka2 from autophagosomes (*Dagdas et al., 2016*). Interestingly, both PexRD54 and Joka2 preferably bind and stimulate formation of autophagosomes marked by potato ATG8CL over ATG8IL, highlighting the selective nature of the process (*Dagdas et al., 2016*). However, the fate of PexRD54 and Joka2 labelled autophagosomes during pathogen attack remains to be elucidated.

## Results

### ATG8CL-autophagosomes localise to haustoria in infected plant cells

To investigate subcellular dynamics of autophagy during infection, we first visualized transiently expressed GFP:ATG8CL in *N. benthamiana* leaf epidermal cells during *P. infestans* infection. GFP:ATG8CL labelled autophagosomes frequently accumulated around the haustoria (73% of observations, $N$ = 60) labelled by the EHM marker REM1.3 (*Figure 1A*), whereas in uninfected cells ATG8CL labelled randomly distributed puncta and the central vacuole (*Figure 1—figure supplement 1A–B*). We previously showed that the EHM could be discriminated from the cytosol and adjacent vacuolar membrane (tonoplast) (*Bozkurt et al., 2015*). Time-lapse microscopy imaging of haustoriated cells in which the tonoplast and EHM are slightly parted away from each other revealed that perihaustorial ATG8CL-autophagosomes with varying size and shape remain tightly associated with the EHM (*Figure 1—figure supplement 2A–C* and *Video 1*). Unlike GFP:ATG8CL, autophagy deficient GFP:ATG8CLΔ mutant failed to accumulate around the haustoria (*Figure 1B*). GFP:ATG8CLΔ appeared randomly distributed as puncta that occurred around haustoria in only 14% of the observations ($N$ = 51) and did not associate with the EHM (*Figure 1B*, *Figure 1—figure supplement 1C*). The GFP control construct showed only diffuse cytoplasmic signal and did not label any perihaustorial puncta (0% $N$ = 20) (*Figure 1C*). To test whether other autophagosomes are targeted toward the haustoria, we investigated subcellular localisation of ATG8IL labelled autophagosomes. Unlike frequently observed GFP:ATG8CL puncta (73% of observations, $N$ = 60) that is abundantly present around the haustoria (*Figure 1A*), GFP:ATG8IL appeared in only 28% the imaged haustoria ($N$ = 65), typically with no more than a few puncta (*Figure 1D*). Taken together, these results demonstrate that during pathogen infection ATG8CL-autophagosomes are selectively directed toward the pathogen interface.

### Accumulation of perihaustorial ATG8CL-autophagosomes is dependent on the core autophagy machinery

To determine the extent to which the core autophagy machinery contributes to the formation of perihaustorial ATG8CL-puncta, we employed RNA interference (RNAi) to knockdown gene

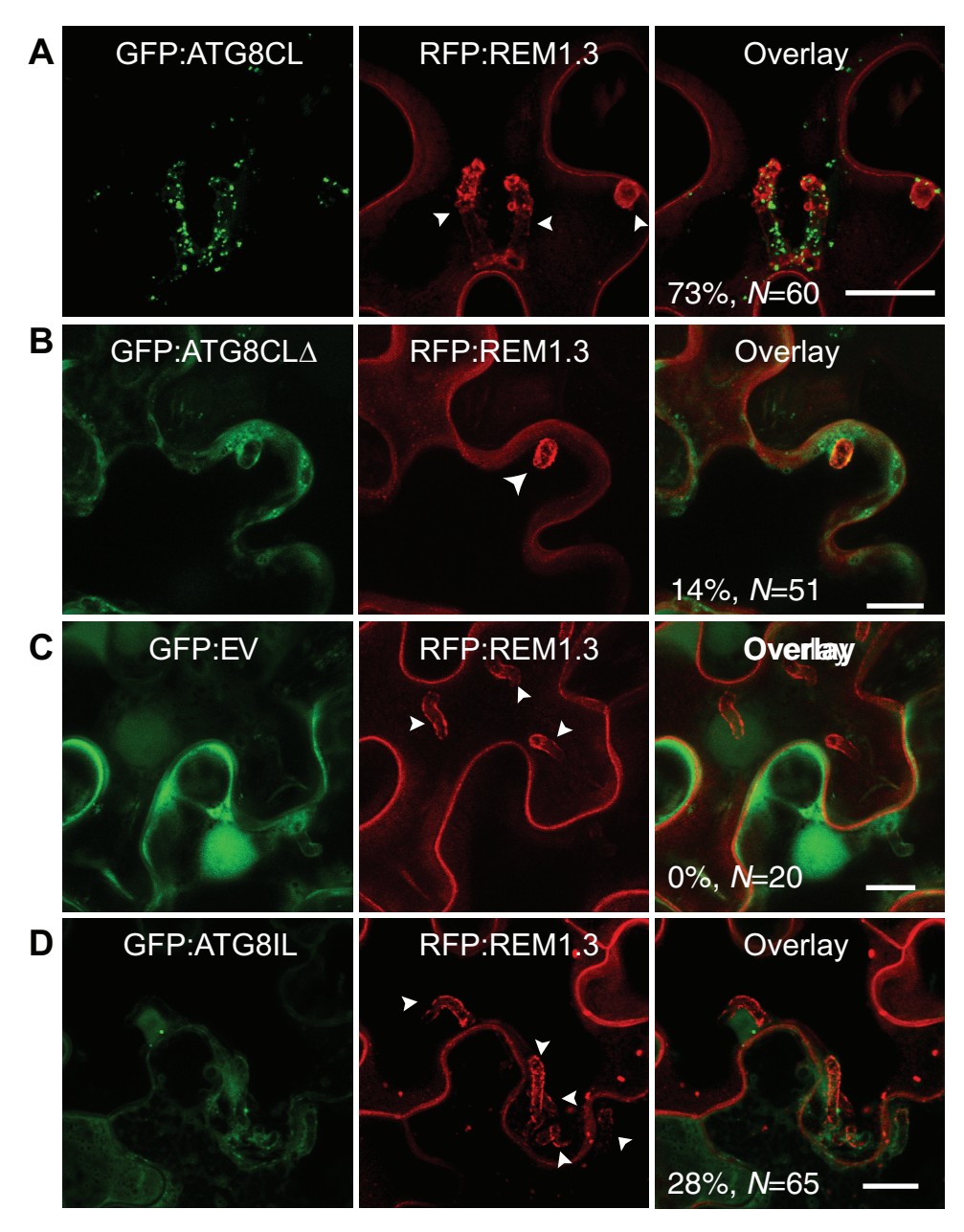

**Figure 1.** ATG8CL-autophagosomes accumulate around the haustoria. GFP:ATG8CL, GFP:ATG8CLΔ, GFP:EV (empty vector) or GFP:ATG8IL are co-expressed with the EHM marker RFP:REM1.3 via agroinfiltration in *N. benthamiana* leaves infected with *P. infestans*. Confocal laser scanning microscopy (CLSM) was used to monitor the autophagosomes in haustoriated cells 3–4 days post infection (dpi). (A) GFP:ATG8CL frequently showed perihaustorial puncta whereas (B) autophagy deficient GFP:ATG8CLΔ appeared as randomly distributed puncta, which failed to accumulate around haustoria (C) GFP:EV did not show any punctate localisation and only labelled perihaustorial cytoplasm (D) GFP:ATG8IL, a divergent member of the ATG8 family, remained mostly cytoplasmic and rarely labelled perihaustorial puncta. Multiple optical sections that fully cover the haustoria are obtained to monitor perihaustorial puncta. Images shown are maximal projections of 16, 10, 15, and 11 frames with 1 μm steps for the top, upper middle, lower middle and bottom rows, respectively. Arrowheads point to haustoria. Scale bars, 10 μm.

DOI: https://doi.org/10.7554/eLife.37476.002

The following figure supplements are available for figure 1:

**Figure supplement 1.** ATG8CL traffics vacuole in uninfected cells.

*Figure 1 continued on next page*

*Figure 1 continued*

DOI: https://doi.org/10.7554/eLife.37476.003

**Figure supplement 2.** ATG8CL-autophagosomes associate with the EHM.

DOI: https://doi.org/10.7554/eLife.37476.004

expression of the core ATG components *ATG4* or *ATG9* (*Lamb et al., 2013*). We quantified haustoria associated with GFP:ATG8CL puncta upon silencing of *ATG4* or *ATG9*, both of which mediate autophagosome biogenesis and maturation. We observed a notable reduction in frequency of perihaustorial ATG8CL-autophagosomes when *ATG4* (31.6% ± 6.2) or *ATG9* (39.3 ± 5.2%) is silenced compared to negative control (*GUS* gene silencing, 63 ± 1.5%) (*Figure 2*, *Figure 2—figure supplement 1*). These results indicate that the core autophagy machinery is required for perihaustorial accumulation of ATG8CL labelled autophagosomes.

## Defense-related selective autophagy mediated by Joka2 is diverted towards haustoria

To investigate focal accumulation of defense-related autophagy components at the pathogen/host interface, we investigated subcellular localisation of Joka2-autophagosomes in haustoriated cells. Joka2 is a modular protein with multiple domains including an N-terminal Phox and Bem1 (PB1) domain, central zinc finger (ZZ) and NBR1 domains followed by two C-terminal ubiquitin-associated domains (UBA) that flank an AIM (*Zientara-Rytter and Sirko, 2014*). PB1 and ZZ domains are implicated in self-oligomerisation and protein-protein interactions, whereas UBA domains and the AIM bridge Joka2 to the autophagic machinery and the ubiquitinated cargo (*Figure 3A*). Autophagosomes labelled by the full length Joka2 fused to BFP (Joka2:BFP) accumulated around the haustoria at high frequency (92%, N = 50) and localised to the EHM, unlike the BFP vector control (*Figure 3B* and *Figure 3—figure supplement 1A–B*). Intriguingly, Joka2$^{AIM}$:BFP mutant also labelled perihaustorial puncta, although at lower frequency (74%, N = 42) compared to Joka2:BFP (*Figure 3C*). Furthermore, a Joka2 truncate lacking the PB1 and ZZ domains, but retaining the ubiquitin binding and ATG8 interacting motifs (Joka2$^{\Delta1-487}$) showed distribution similar to the BFP vector control and failed to accumulate at the pathogen interface (1%, N = 72) (*Figure 3D,E*). This was not due to reduced protein stability as Joka2:BFP$^{\Delta1-487}$ accumulated at similar protein levels compared to Joka2:BFP (*Figure 3—figure supplement 2*). These findings suggest that Joka2-ATG8 interaction is not sufficient for Joka2's recruitment to the perihaustorial puncta and Joka2's oligomerisation and/or association with other proteins mediated by PB1 and ZZ domains are critical for its haustorial accumulation.

To validate that Joka2 localizes to perihaustorial ATG8CL autophagosomes, we co-expressed Joka2:BFP, Joka2$^{AIM}$:BFP or BFP:EV with GFP:ATG8CL in haustoriated *N. benthamiana* cells marked by RFP:REM1.3. Joka2:BFP fluorescent signal fully overlapped with GFP:ATG8CL labelled perihaustorial autophagosomes (100%, N = 140) unlike the BFP:EV (0%, N = 20) indicating Joka2 localizes to perihaustorial ATG8CL autophagosomes (*Figure 3—figure supplement 3A–C*). Surprisingly, Joka2:BFP also labelled puncta that did not show any GFP:ATG8CL

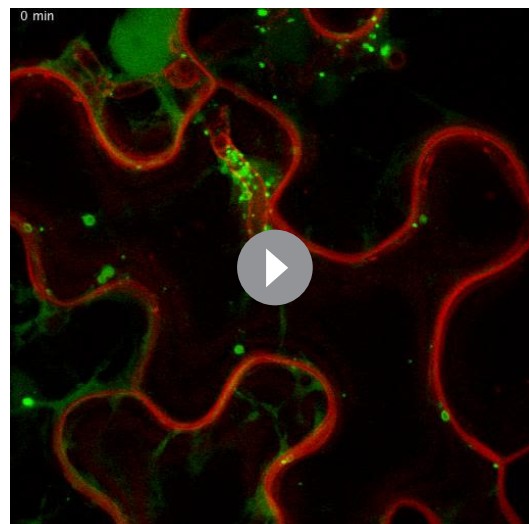

**Video 1.** ATG8CL-autophagosomes accumulate around the haustoria and remain associated with the EHM. GFP:ATG8CL is co-expressed with the EHM marker RFP:REM1.3 via agroinfiltration in *N. benthamiana* leaves infected with *P. infestans*. Confocal laser scanning microscopy was used to monitor the autophagosomes in haustoriated cells three dpi (days post infection). The movie represents time-lapse of maximal projections of 9 frames with 1.5 µm steps acquired during 15 min (Frame interval: 27 s).

DOI: https://doi.org/10.7554/eLife.37476.005

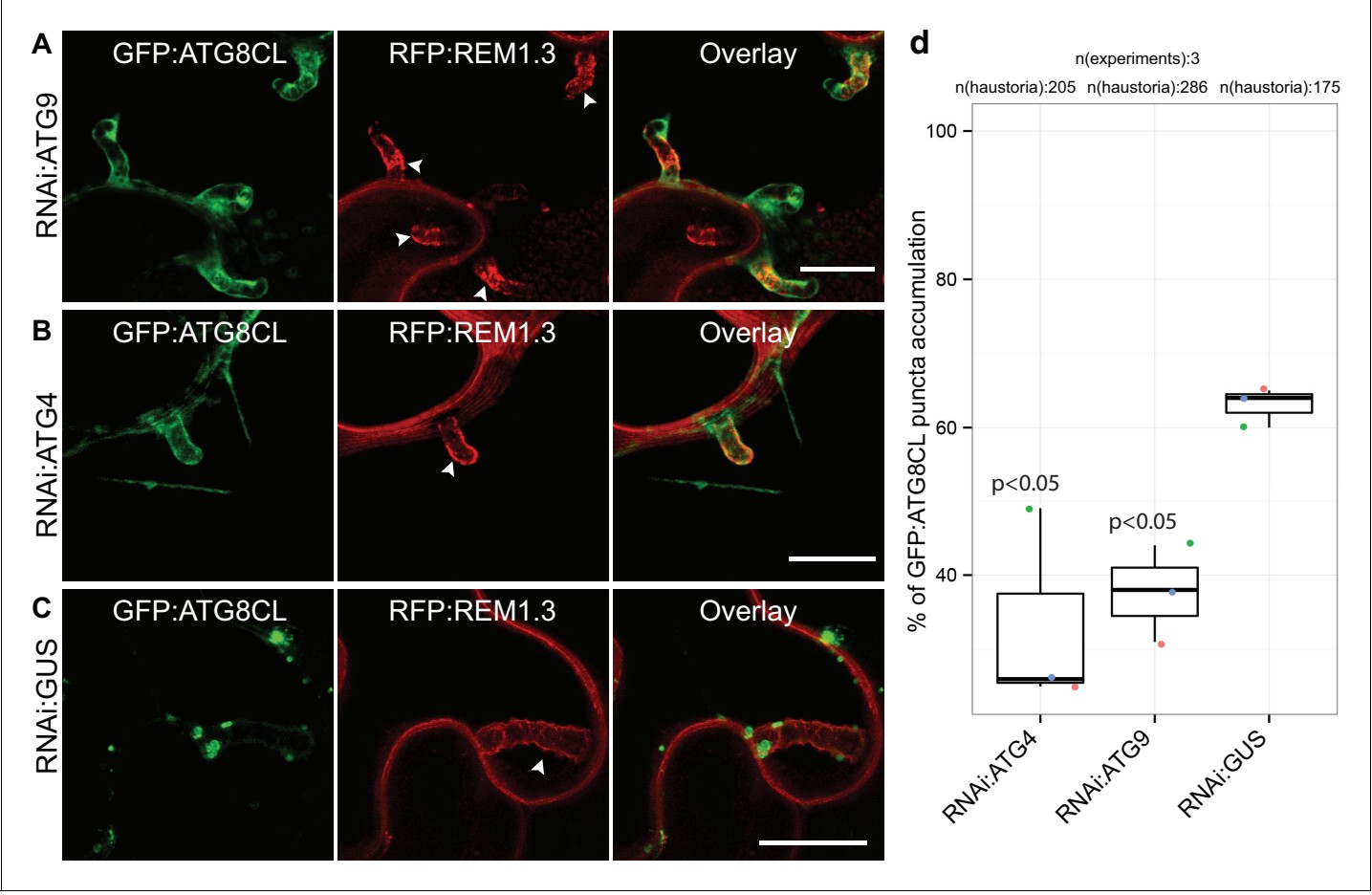

**Figure 2.** ATG4 and ATG9 are required for perihaustorial accumulation of ATG8CL autophagosomes. (A–D) In infected leaf patches, GFP:ATG8CL is co-expressed with RFP:REM1.3 in the presence of hairpin RNAi constructs targeting *ATG4*, *ATG9* or control *GUS*. CSLM analyses of three independent experiments revealed that perihaustorial accumulation of autophagosomes are significantly reduced when *ATG9* (A) or *ATG4* (B) is silenced compared to *GUS* (C) silencing. Images shown are maximal projections of 12, 10 and 9 frames with 1 μm steps from top to bottom rows, respectively. Arrowheads point to haustoria. Scale bars, 10 μm. Images were obtained 3–4 dpi. (D) Quantification of perihaustorial ATG8CL-puncta upon *ATG9, ATG4 or GUS* silencing.

DOI: https://doi.org/10.7554/eLife.37476.006

The following figure supplement is available for figure 2:

**Figure supplement 1.** Validation of ATG4 and ATG9 knockdowns.

DOI: https://doi.org/10.7554/eLife.37476.007

fluorescence suggesting that Joka2 also labels compartments that are not ATG8CL-autophago-somes (*Figure 3—figure supplement 3A*). Consistent with this, Joka2[AIM]:BFP produced fluorescence signal at discrete puncta that rarely coincided with perihaustorial ATG8CL-autophagosomes (19%, $N = 37$) (*Figure 3—figure supplement 3B*). Like mammalian autophagy cargo receptors, Joka2 forms oligomers (*Zientara-Rytter and Sirko, 2014*), and this most likely accounts for recruitment of Joka2[AIM]:BFP to ATG8CL-autophagosomes.

Based on these observations, we hypothesized that diversion of Joka2 mediated autophagy towards haustoria would lead to a decrease in vacuolar degradation of Joka2. To test this, we measured autophagic flux in infected leaves. We observed that, similar to ATG8CL, Joka2 protein levels were higher in infected leaves compared to the mock infected leaves (*Figure 3—figure supplement 4*). These results further illustrate that ATG8CL/Joka2 mediated antimicrobial autophagy is targeted to the haustorial interface.

Previously we have shown that overexpression of Joka2 restricts pathogen colonisation (*Dagdas et al., 2016*). To test if focal accumulation of Joka2 is important for its antimicrobial

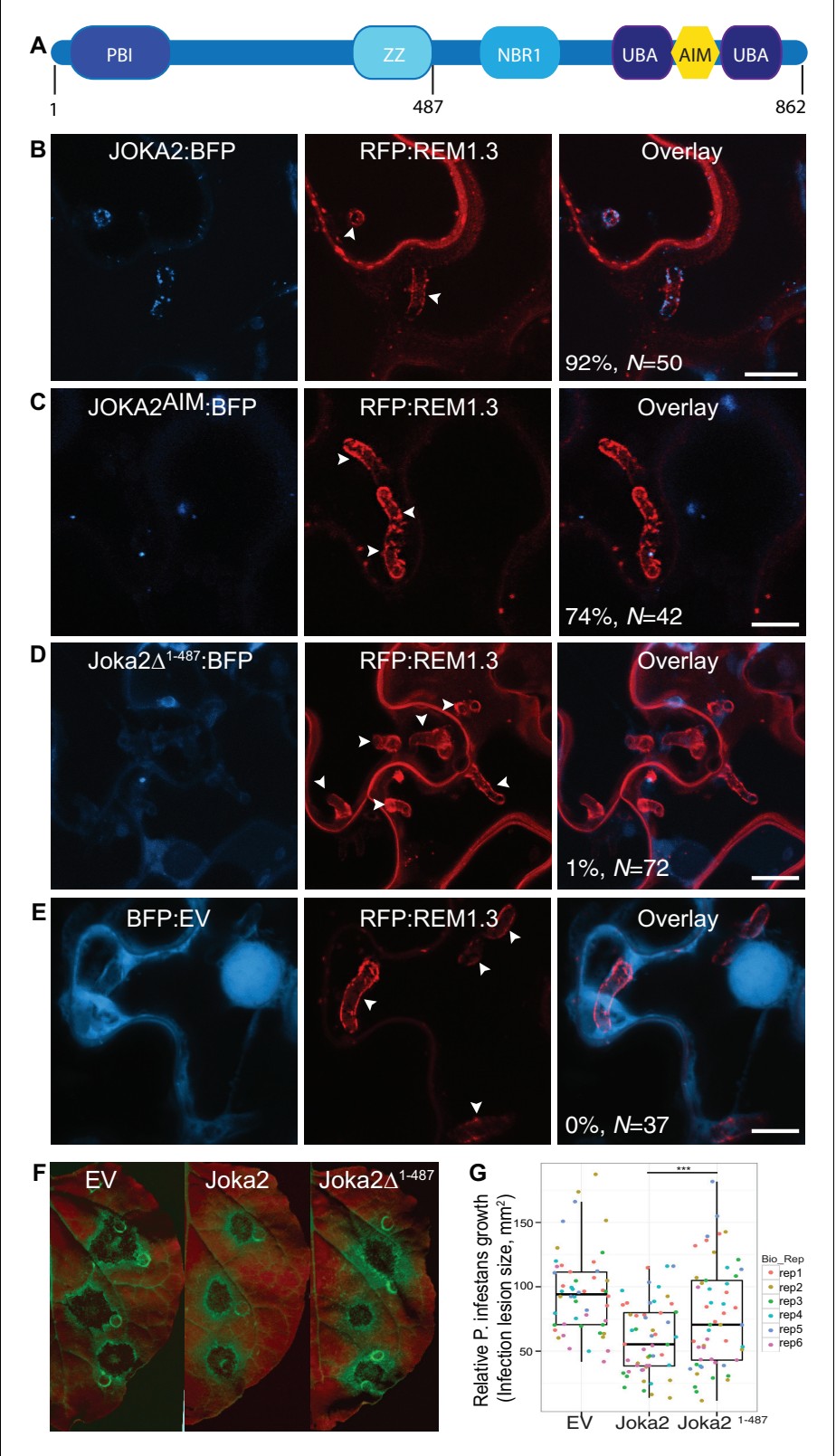

**Figure 3.** Joka2-mediated antimicrobial autophagy is directed toward the haustoria. (**A**) Joka2 domain architecture (**B–E**) Confocal microscopy of *P. infestans* infected *N. benthamiana* leaf epidermal cells expressing Joka2:BFP, Joka2<sup>AIM</sup>:BFP mutant, Joka2<sup>Δ1-487</sup>:BFP or BFP:EV control. Both Joka2:BFP (top panel) and Joka2<sup>AIM</sup>:BFP (mid panel) displayed perihaustorial puncta although the frequency of the later was much lower. Consistently, *Figure 3 continued on next page*

*Figure 3 continued*

Joka2$^{\Delta1-487}$:BFP mainly showed cytoplasmic distribution similar to BFP:EV control and rarely marked perihaustorial puncta (1.3% of imaged haustoria). Images shown are maximal projections of 10, 14, 10 and 7 frames with 1 μm steps from top to bottom rows, respectively. Arrowheads point to haustoria. Scale bars, 10 μm. Images were obtained 3–4 dpi. (F–G) Full length Joka2 enhances disease resistance against *P. infestans,* whereas Joka2$^{\Delta1-487}$, which does not accumulate around haustoria (D), only provides partial resistance. (F) *N. benthamiana* leaves expressing Joka2, Joka2$^{\Delta1-487}$ and empty vector (EV) control were infected with *P. infestans* and pathogen growth was determined by measuring infection lesion size eight days post-inoculation. (G) Categorical scatter plots illustrate infection lesion size of 8–10 infections sites from six independent biological replicates pointed out by six different colours. Welch Two Sample t-test revealed a significant difference (***p=0.0126) in disease resistance conferred by Joka2 compared to Joka2$^{\Delta1-487}$.

DOI: https://doi.org/10.7554/eLife.37476.008

The following figure supplements are available for figure 3:

**Figure supplement 1.** Joka2/ATG8CL-autophagosomes associate with the EHM.
DOI: https://doi.org/10.7554/eLife.37476.009
**Figure supplement 2.** Joka2$^{\Delta1-487}$ is stably expressed.
DOI: https://doi.org/10.7554/eLife.37476.010
**Figure supplement 3.** Joka2 localizes to ATG8CL-autophagosomes around the haustoria.
DOI: https://doi.org/10.7554/eLife.37476.011
**Figure supplement 4.** Joka2 degradation slows down during *P. infestans* infection.
DOI: https://doi.org/10.7554/eLife.37476.012

function, we infected Joka2 and Joka2:BFP$^{\Delta1-487}$ overexpressing leaves with *P. infestans.* In contrast to leaf patches expressing the full length Joka2, which can accumulate at the perihaustorial autophagosomes (*Figure 3B*), leaves expressing Joka2$^{\Delta1-487}$ conferred only mild resistance (*Figure 3F–G*). Considered together with the cell biological analyses of Joka2 in haustoriated cells, infection assays suggest that diversion of plant antimicrobial autophagy towards pathogen interface is critical to limit *P. infestans* infection. Thus, the pathogen needs to subvert the biogenesis of antimicrobial Joka2/ATG8CL compartments and/or neutralize their defense related function at the host-pathogen interface.

## *Phytophthora infestans* effector PexRD54 accumulates at Haustoria

To explore focal subversion of autophagic defense responses during infection, we set out to determine the subcellular localisation of the *P. infestans* effector PexRD54 in infected cells. In haustoriated cells, GFP:PexRD54 frequently labelled perihaustorial puncta (70%, *N* = 36) (*Figure 4A*). However, similar to the GFP control (0%, *N* = 22), we rarely detected any puncta around the haustoria labelled by GFP:PexRD54$^{AIM2}$ (4%, *N* = 71, *Figure 4B–C*). This suggests that ATG8CL binding is critical for recruitment of PexRD54 to the perihaustorial autophagosomes. To determine whether PexRD54 labelled vesicles are ATG8CL-autophagosomes, we co-expressed BFP:PexRD54 with GFP:ATG8CL in haustoriated *N. benthamiana* cells. We observed a full overlap between the two punctate fluorescent signals across the EHM, in contrast to BFP:PexRD54$^{AIM2}$ and BFP:EV negative controls (100%, *N* = 73 for PexRD54, 14%, *N* = 35 for PexRD54$^{AIM2}$ and 0%, *N* = 29 for EV control) (*Figure 4D–G*). Although hardly observed (5/35, 14%), detection of perihaustorial PexRD54$^{AIM2}$ labelled autophagosomes suggests that this mutant can still weakly associate with ATG8CL in vivo or forms higher order molecular complexes with the host autophagy machinery. Altogether, these findings demonstrate that the ATG8CL selective autophagy pathway that is targeted by *P. infestans* is diverted to the haustorial interface.

## Host-microbe interface is a hotspot for autophagosome biogenesis

We next investigated the origin of perihaustorial autophagosomes by testing whether they are synthesized at the host-pathogen interface or traffic to these sites following their biogenesis in other subcellular regions. A recent study has shown that the plant ATG9 homolog, the only transmembrane domain containing ATG protein, localises to autophagosome biogenesis sites and remains in mobile puncta adjacent to mature autophagosomes (*Zhuang et al., 2017*). Hence, we used ATG9:GFP as a marker for phagophore assembly sites (PAS) and monitored its localisation during infection.

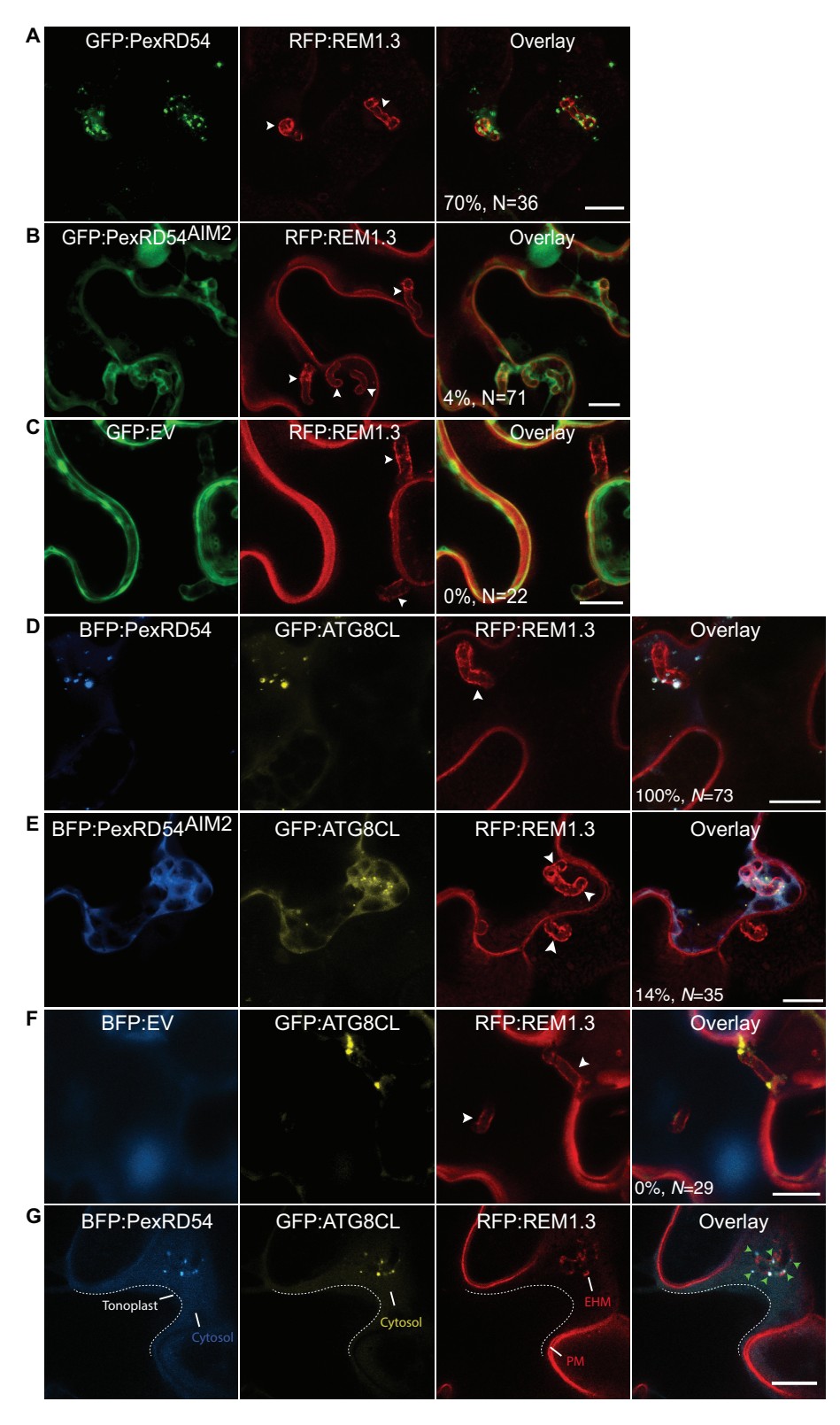

**Figure 4.** PexRD54 accumulates at perihaustorial autophagosomes. (A–C) Confocal images of haustoriated plant cells marked by REM1.3. GFP: PexRD54 showed frequent perihaustorial puncta (A) unlike its ATG8 interaction motif mutant PexRD54^AIM2 (B), which rarely labelled perihaustorial puncta and showed cytoplasmic distribution similar to GFP:EV (C). (D–F) In haustoriated cells marked by RFP:REM1.3, GFP:ATG8CL labelled autophagosomes fully overlapped with perihaustorial BFP:PexRD54 puncta (D). In contrast, BFP:PexRD54^AIM2 mainly remained cytoplasmic and mostly

*Figure 4 continued on next page*

*Figure 4 continued*

did not show perihaustorial puncta that overlap with ATG8CL autophagosomes (**E**), similar to BFP:EV control (**F**). Images shown are maximum projections of 8, 12, 10, 8, 5, and 9 frames with 1 μm steps from top to bottom rows, respectively. White arrowheads point to haustoria. Scale bars, 10 μm. Images are obtained 3–4 dpi. (**G**) PexRD54/ATG8CL labelled autophagosomes accumulate across the EHM. Single focal plane CLSM images show regions where the EHM and the tonoplast (dotted line) are parted away and the cytosol is no longer a thin layer between the two membranes. Autophagosomes co-labelled by BFP:PexRD54 and GFP:ATG8CL associate with the EHM marked by RFP:REM1.3. Green arrowheads in overlay panel point to BFP:PexRD54 labelled autophagosomes that associate with the EHM marked by RFP:REM1.3. Scale bar, 10 μm.

DOI: https://doi.org/10.7554/eLife.37476.013

Transient expression of ATG9:GFP in haustoriated cells revealed that ATG9 is ubiquitously found at the perihaustorial puncta (in >92% of imaged haustoria) neighbouring ATG8CL and Joka2 labelled perihaustorial autophagosomes with a partial yet clear overlap (*Figure 5A–C*). This finding suggests that in addition to accommodating mature ATG8CL autophagosomes, perihaustorial compartments are hotspots for autophagosome biogenesis. Besides, we noted a substantial increase in the frequency of haustoria that associate with ATG8CL autophagosomes in infected plant cells overexpressing ATG9:GFP (90%, *N* = 120) compared to cells expressing GFP as a control (67%, *N* = 105) (*Figure 5—figure supplement 1*). Accumulation of ATG9 around the haustoria and its boosting effect on the perihaustorial autophagosomes prompted us to test its role in immunity. For this, we infected ATG9:GFP and GFP:EV expressing leaves with *P. infestans* (*Figure 5D*). Strikingly, we repeatedly observed that increasing ATG9 protein levels led to a significant drop in *P. infestans* infection (p=0.0035, six biological replicates). Overall these results suggest selective autophagy functions as a focal immune response against *P. infestans*. (*Figure 5D–E*).

## Discussion

In this study, we combined high-resolution microscopy with functional genetic analysis to monitor the course of defense related autophagy in *N. benthamiana* cells during *P. infestans* infection. We show that autophagosomes labelled by the core autophagy protein ATG8CL and the plant autophagy cargo receptor Joka2 are diverted to the EHM, and accumulation of Joka2 at this interface is critical for its antimicrobial activity (*Figures 1–3*). Moreover, *P. infestans* RXLR effector PexRD54, which functions as a competitive inhibitor of Joka2, also accumulated across the EHM (*Figure 4*). These findings suggest that previously discovered antagonistic interaction between PexRD54 and Joka2 principally takes place across the EHM, where PexRD54 depletes Joka2 from ATG8CL-autophagosomes to undermine antimicrobial autophagy. A similar antagonistic interaction has recently been reported in plasmodium infected hepatocytes, in which a plasmodium virulence factor counteracts functioning of the mammalian xenophagy receptor p62 on the parasitophorous vacuole membrane that accommodates the intracellular plasmodium (*Real et al., 2018*). These findings highlight convergent evolution of autophagy related host defenses, guided by autophagy cargo receptors, targeted towards the invading plant and animal pathogens. Furthermore, our results illustrate that adapted plant pathogens deploy effector proteins to remodel processes taking place at the perimicrobial host membranes and antagonize the focal immune responses deployed by the host to destroy the invaders.

Recruitment of ATG9 labelled vesicles to perihaustorial region suggests the pathogen interface serves a scaffold for autophagosome formation and defense related autophagy responses (*Figure 5*). This would not only eliminate the unnecessary energy spent to transport these defense-related spherical bodies towards the EHM, but more importantly, minimize the time required for their effective deployment at this interface. This is reminiscent of the antibacterial autophagy responses mounted by mammalian cells against *Salmonella Typhimurium* (*Randow et al., 2013*). Future studies focusing on dissecting the components of the perihaustorial autophagosome biogenesis machinery should provide insights into how antimicrobial autophagy is accurately guided towards pathogen invasion sites.

Surprisingly, ATG8 gene family has expanded and diversified to different degrees in plant lineages (*Kellner et al., 2017*). The expansion of plant ATG8s appears to have occurred early in evolution, and each plant family carries a unique set of ATG8 isoforms that have diversified over millions of years (*Kellner et al., 2017*). Thus, it is possible that in different plant families different ATG8

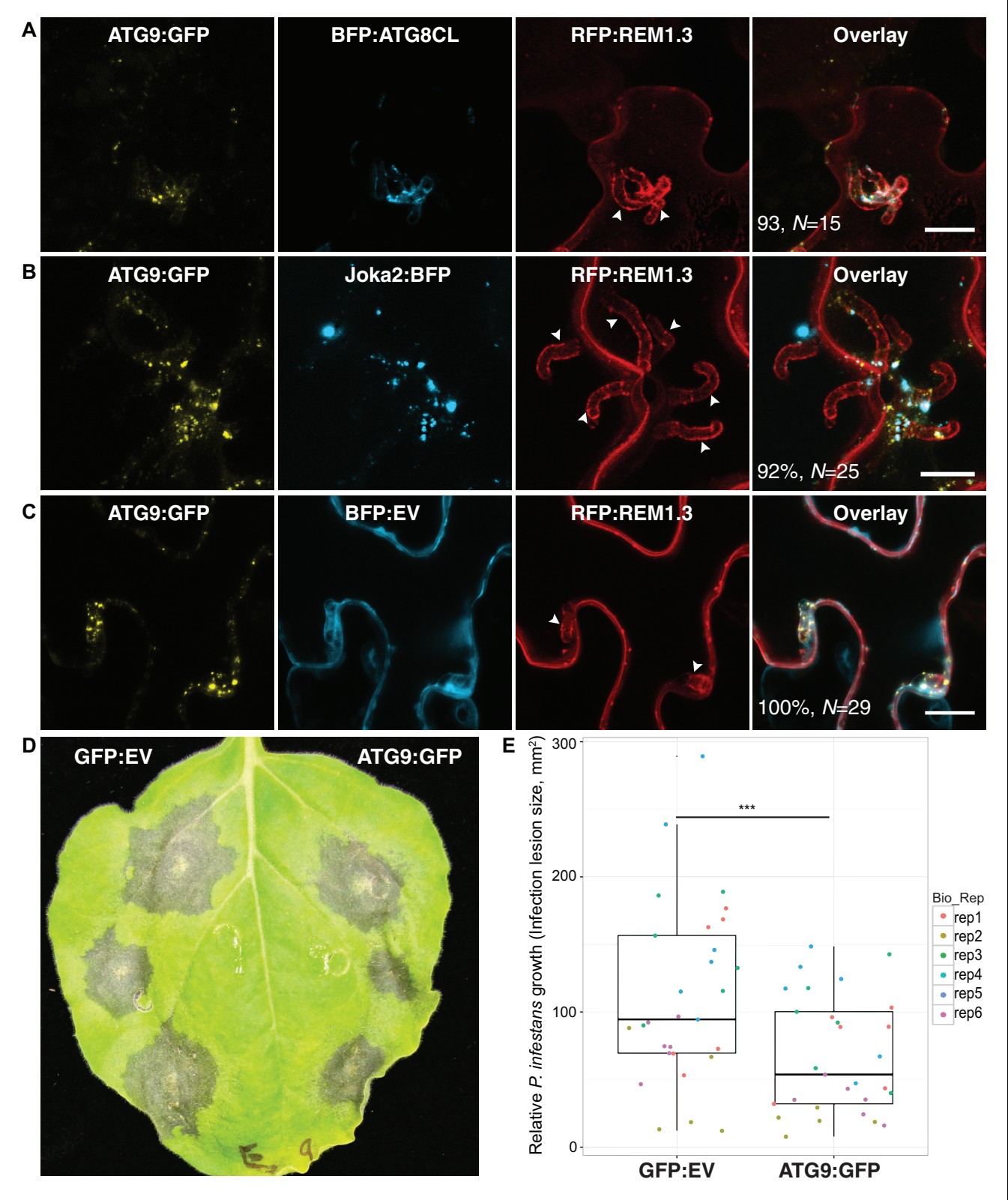

**Figure 5.** ATG9 accumulates around the haustoria and contributes to immunity. (**A–C**) ATG9 localizes to ATG8CL-autophagosomes around the haustoria. ATG9:GFP labelled puncta frequently observed around the haustoria (>%92 of the imaged haustoria in (**A**), (**B, C**) which partially overlapped with the perihaustorial autophagosomes marked by BFP:ATG8CL and Joka2:BFP (**A, B**), but not with BFP:EV control (**C**). Images shown are maximal projections of 16, 15 and 7, frames with 1 μm steps from top to bottom panels, respectively. Arrowheads point to haustoria. Scale bars, 10 μm. Images

*Figure 5 continued on next page*

*Figure 5 continued*

are obtained three dpi.(**d–e**) ATG9 overexpression enhances disease resistance against *P. infestans*. (**D**) *N. benthamiana* leaves expressing on each half either ATG9:GFP or GFP empty vector (GFP:EV) control were infected with *P. infestans* and pathogen growth was determined by measuring infection lesion size eight days post-inoculation. (**E**) Categorical scatter plots illustrate infection lesion size of 6 infections sites (except rep2 with five infection sites) from six independent biological replicates pointed out by six different colours. ATG9 significantly (\*\*\*$p<0.01$) enhanced disease resistance against *P. infestans*.

DOI: https://doi.org/10.7554/eLife.37476.014

The following figure supplement is available for figure 5:

**Figure supplement 1.** ATG9 overexpression enhances the frequency of haustoria that associate with ATG8CL-autophagosomes.

DOI: https://doi.org/10.7554/eLife.37476.015

isoforms are specialized to carry out defense-related tasks. A phylogenetic analysis of solanaceous plants revealed family-specific groups of ATG8 members that form four distinct clades (*Kellner et al., 2017*). Unlike ATG8CL, another solanaceous ATG8 clade member, ATG8IL, does not accumulate at the haustorial interface during *P. infestans* infection. Thus, not all autophagic trafficking components are diverted to the haustorial interface. Autophagosomes labelled by ATG8IL do not respond to pathogen infection and remain cytosolic, further highlighting the specialization of ATG8 isoforms in plants that was proposed by Kellner et al (*Kellner et al., 2017*). (*Figure 1*). These findings reveal a novel trafficking route from the cytoplasm to the pathogen interface and expands our understanding of the biogenesis of the EHM (*Bozkurt et al., 2015*, *2014*; *Lu et al., 2012*). The emerging view is that the EHM is formed by the redirection of different endomembrane trafficking pathways, notably the late endocytic pathway and ATG8CL-mediated selective autophagy.

We hypothesize that the PexRD54-ATG8CL autophagosomes carry a distinct cargo that substitutes the defense related cargo to redirect molecules towards the pathogen. Such pathogen modified double-layered autophagosomes could fuse with the EHM discharging single layered vesicles into the extrahaustorial matrix. Fusion of autophagosomes with the EHM could provide a membrane source for EHM biogenesis and may account for the extracellular vesicles (EVs) that have been recently reported in several host-microbe interfaces (*Deeks and Sánchez-Rodríguez, 2016*; *Rutter and Innes, 2017*). Thus, PexRD54 may orchestrate the recruitment of host cargo for delivery as EVs to the host-pathogen interface. Further studies are required to determine the precise mechanisms that govern autophagosome biogenesis at the haustorial interface and its impact in pathogenicity. Moreover, identifying the nature of the autophagosome cargo sequestered by PexRD54 and Joka2 will further expand our understanding of the role of selective autophagy in host-microbe interactions.

## Materials and methods

### Molecular cloning and plasmid constructs

GFP:ATG8CL, GFP:ATG8CLΔ, GFP:ATG8IL, GFP:PexRD54, GFP:PexRD54^AIM, GFP:EV and RFP:REM1.3 constructs were previously described (*Bozkurt et al., 2015*). All other blue fluorescent protein (BFP) fusion constructs were generated in this study. The vector for N-terminal BFP fusion was derived from pK7WGF2 plasmid (*Karimi et al., 2002*) by excising a fragment from the backbone with EcoRV digestion then replacing it with a custom synthesized fragment containing tagBFP sequence followed by linker sequence (GGATCTGCTGGATCTGCTGCTGGATCTGGAGAATTT) and EcoRV restriction site (where the gene of interest will be inserted) (Eurofins Genomics). Similarly, the vector for C-terminal BFP fusion was also derived from pK7WGF2 plasmid but by inserting PCR fragments containing EcoRV restriction site (where the gene of interest will be inserted) followed by linker sequence (GGATCTGCTGGATCTGCTGCTGGATCTGGAGAATTTGGATCA) and tagBFP sequence amplified from N-terminal BFP fusion vector using primer pairs GA_35 s_F with Cterm_BFP_Prom_R and Cterm_BFP_F with Cterm_BFP_R. Then, ATG9:GFP, BFP:PexRD54, BFP:PexRD54^AIM2, Joka2:BFP and Joka2^AIM:BFP, Joka2^Δ1-487:BFP constructs were generated by Gibson assembly of each gene PCR fragment into EcoRV digested GFP/BFP vectors (N-terminal fusion for PexRD54 and PexRD54^AIM, C-terminal fusion for ATG9, Joka2^Δ1-487, Joka2 and Joka2^AIM). All genes except ATG9, which was amplified from *N. benthamiana* cDNA, were amplified from existing

constructs previously described (*Bozkurt et al., 2015*), using primer pairs GA_RD54_F with GA_RD54_R for PexRD54, GA_RD54_F with GA_LIR2_R for PexRD54$^{AIM}$ and GA_Joka2_BFP_F with GA_Joka2_BFP_R for both Joka2, Joka2$^{AIM}$, and GA_Joka2$^{Δ1-487}$_F, and GA_ATG9_F with GA_ATG9_R. Silencing constructs for *ATG4* and *ATG9* were amplified using the primer combinations hpATG4_F/hpATG4_R and hpATG9_F/hpATG9_R and cloned into the pRNAiGG vector, following the protocol from Pu Yan et al. (*Yan et al., 2012*). All primers used in this study are listed in *Supplementary file 1*.

## *ATG4* and *ATG9* silencing assays

A BLASTP search of *ATG4* and *ATG9* against *N. benthamiana* proteins in the Sol Genomics database revealed one coding region for *ATG4* (Niben101Scf02450g03007.1) and two homologs of *ATG9*, referred to here as *ATG9A* and *ATG9B* (Niben101Scf00114g00010.1 and Niben101Scf08519g00001.1). A hairpin RNAi construct targeting a conserved region in ATG9a/b was designed to silence both *ATG9* homologs. Silencing of *ATG4* and *ATG9 was* verified using RT-PCR. Total RNA was extracted using GeneJET Plant RNA purification Mini Kit (Thermo Scientific). 2 μg of RNA was used for cDNA synthesis using SuperScript IV Reverse Transcriptase (Invitrogen). RT was performed with the following conditions: 50 min at 55°C followed by 20 min at 70°C. Primers pairs used for cDNA amplification were RT_ATG4_F/RT_ATG4_R, RT_ATG9A_F/RT_ATG9A_R, and RT-ATG9B F/RT-ATG9B R. *GAPDH* was used to normalize transcript abundance. All primers used in this study are listed in *Supplementary file 1*.

## Confocal microscopy

Imaging was performed using Leica SP5 resonant inverted confocal microscope (Leica Microsystems) using 63x water immersion objective. All microscopy analyses were carried out on live leaf tissue 3–4 days after agroinfiltration. Leaf discs of *N. benthamiana* were cut and mounted onto Carolina observation gel (Carolina Biological Supply Company) to minimize the damage. Specific excitation wavelengths and filters for emission spectra were set as described previously (*Koh et al., 2005*). BFP, GFP and RFP probes were excited using 405, 488 and 561 nm laser diodes and their fluorescent emissions detected at 450–480, 495–550 and 570–620 nm, respectively. Sequential scanning between lines was done to avoid spectral mixing from different fluorophores and images acquired using multichannel. Maximum intensity projections of Z-stack images were presented in each figure. Z-stack sections were processed to enhance image clarity, sections that caused blurriness (top and bottom ones), were removed for generation of maximum intensity projections. Image analysis was performed using ImageJ (1.50 g) and Adobe Photoshop (CS6).

## Transient gene-expression assays in *N. benthamiana*

Transient gene-expression was performed *in planta* by infiltration of leaves of 3–4 week old *N. benthamiana* with cultures of *Agrobacterium tumefaciens* GV3101 strain carrying T-DNA constructs, as previously described (*Bozkurt et al., 2011*). Transient co-expression assays were carried out by mixing equal ratios of *A. tumefaciens* carrying the plant expression constructs in agroinfiltration medium [10 mM MgCl$_2$, 5 mM 2-(N-morpholine)-ethanesulfonic acid (MES), pH 5.6] to achieve a final OD$_{600}$ of 0.2.

## Biological material

*N. benthamiana* plants were grown and maintained in a greenhouse with high light intensity (16 hr light/8 hr dark photoperiod) at 22–24°C. *P. infestans* strain 88069 cultures (*van West et al., 1998*) were grown and maintained on rye sucrose agar medium at 18°C in the dark for 12–14 days, as described elsewhere (*Song et al., 2009*) prior to use for infection of *N. benthamiana*. Zoospores were released from sporangia by addition of cold water and incubation at 4°C for 90 min adjusting dilution to 50,000 spores/ml. Infection of agroinfiltrated leaves was carried out by addition of 10 μl droplets containing zoospores as described previously (*Song et al., 2009*; *Saunders et al., 2012*) with the exception that infection was carried out on attached leaves, incubating inoculated plants in humid growth chambers.

## Acknowledgements

We thank members of the Bozkurt and Kamoun Labs for helpful suggestions. This project was funded by the Gatsby Charitable Foundation, European Research Council (ERC), and Biotechnology Biological Sciences Research Council (BBSRC).

## Additional information

### Funding

| Funder | Grant reference number | Author |
|---|---|---|
| Gatsby Charitable Foundation | | Yasin F Dagdas<br>Khaoula Belhaj<br>Sophien Kamoun<br>Tolga O Bozkurt |
| European Research Council | | Yasin F Dagdas<br>Khaoula Belhaj<br>Sophien Kamoun |
| Biotechnology and Biological Sciences Research Council | BB/M002462/1 | Pooja Pandey<br>Yasin Tumtas<br>Nattapong Sanguankiattichai<br>Cian Duggan<br>Alexandre Y Leary<br>Maria E Segretin<br>Mauricio P Contreras<br>Zachary Savage<br>Tolga O Bozkurt |

The funders had no role in study design, data collection and interpretation, or the decision to submit the work for publication.

### Author contributions

Yasin F Dagdas, Conceptualization, Resources, Data curation, Formal analysis, Investigation, Visualization, Methodology, Writing—original draft, Writing—review and editing; Pooja Pandey, Conceptualization, Resources, Data curation, Formal analysis, Supervision, Validation, Investigation, Visualization, Methodology, Writing—original draft, Writing—review and editing; Yasin Tumtas, Resources, Formal analysis, Visualization, Methodology, Writing—review and editing; Nattapong Sanguankiattichai, Resources, Formal analysis, Investigation, Visualization, Methodology, Writing—review and editing; Khaoula Belhaj, Conceptualization, Validation, Visualization, Methodology; Cian Duggan, Resources, Data curation, Investigation, Visualization, Methodology, Writing—review and editing; Alexandre Y Leary, Formal analysis, Investigation, Visualization, Writing—review and editing; Maria E Segretin, Supervision, Validation, Investigation, Visualization, Methodology, Writing—review and editing; Mauricio P Contreras, Resources, Investigation, Visualization, Methodology; Zachary Savage, Resources, Formal analysis, Investigation, Visualization; Virendrasinh S Khandare, Formal analysis, Investigation, Visualization; Sophien Kamoun, Investigation, Visualization, Methodology; Tolga O Bozkurt, Conceptualization, Formal analysis, Supervision, Funding acquisition, Writing—original draft, Project administration, Writing—review and editing

### Author ORCIDs

Yasin F Dagdas http://orcid.org/0000-0002-9502-355X
Pooja Pandey http://orcid.org/0000-0003-3145-7794
Alexandre Y Leary http://orcid.org/0000-0001-7223-3557
Sophien Kamoun https://orcid.org/0000-0002-0290-0315
Tolga O Bozkurt http://orcid.org/0000-0003-0507-6875

### Decision letter and Author response

Decision letter https://doi.org/10.7554/eLife.37476.020
Author response https://doi.org/10.7554/eLife.37476.021

## Additional files

### Supplementary files
• Supplementary file 1. Primers used in this study.
DOI: https://doi.org/10.7554/eLife.37476.016
• Transparent reporting form
DOI: https://doi.org/10.7554/eLife.37476.017

### Data availability
All data generated or analysed during this study are included in the manuscript and supporting files.

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
