## [Decision Letter]

[Editors’ note: a previous version of this study was rejected after peer review, but the authors submitted for reconsideration. The first decision letter after peer review is shown below.]

Thank you for submitting your work entitled "Host autophagosomes are diverted to a plant-pathogen interface" for consideration by *eLife*. Your article has been reviewed by three peer reviewers, and the evaluation has been overseen by a Reviewing Editor and a Senior Editor.

Our decision has been reached after consultation between the reviewers. Based on these discussions and the individual reviews below, we regret to inform you that your work will not be considered further for publication in *eLife*.

There was general agreement that the study is interesting. However, the study falls short of providing enough new mechanistic insights to be suitable for a research advance. Specifically, there are important issues that are not fully substantiated, which raised questions:

-Do the events described in the manuscript happen during infection? (That is, what is the biological relevance during natural infection?)

-Are autophagosomes actually directed toward the pathogen interface under native conditions? Additional microscopy (e.g. electron microscopy) can help address this important issue.

Reviewer #1:

The manuscript by Dagdas et al., is a continuation of research from this group that studies effectors from the oomycete *Phytophthora infestans*. Effector biology, mode action, localization/translocation and importantly, the underlying mechanistic details responsible for this intriguing class of regulators. Previously, this group made an interesting discovery, coupling effector research with another emerging (emerged) area of research that is operative in both plants and animals, namely autophagy. Utilizing an effector from *P. infestans*; (Pex RD54). In mechanistic studies the authors were able to show that this oomycete effector binds to a host protein, to trigger autophagosome formation. Interestingly during infection where autophagosomes usually deliver their cargo, instead *P. infestans* triggers autophagosome formation that results in the depletion of autophagy cargo including Joka2 a selective ATG receptor. How does this work? How is this used to establish compatibility?

Finally, in the current manuscript, data is presented that suggests that Autophagosomes are "diverted" to the pathogen interface and in a resistant response results in starvation. Conclusions include the idea that the pathogen reprograms host selective autophagy purely for the benefit of the pathogen. This a very nice paper in many ways and will likely be a seminal contribution to the field merging Molecular Plant Pathology and autophagic cell death.

1) My main concern is the microscopy. Not being a microscopist … but can the resolution be improved? It is hard to determine what is happening during these interactions. Would TEM help?

Reviewer #2:

This research advance builds upon previous work where the *P. infestans* effector PexRD54 specifically binds to the plant protein ATG8CL to induce autophagosome formation. Here, the authors demonstrate that ATG8CL, but not AGT8IL, localize to haustoria after transient expression in *Nicotiana* plants infected with *P. infestans* using the expression of the remorin protein REM1.3 as a marker for *P. infestans* haustoria. The autophagy cargo receptor Joka2 also accumulates at haustoria upon infection. Previous papers have demonstrated targeted vesicular trafficking to the site of pathogen infection, indicating that trafficking of cellular components to the site of pathogen ingress is a common phenomenon.

The microscopy images are striking and in general the experiments are well performed. However, all the data rely upon transient overexpression in *Nicotiana*. Does this occur during natural infection under native expression?

The biological relevance of this relocalization is also not clear. Identifying the cargo of autophagosomes relocalized to the site of pathogen infection is required to understand the role of selective autophagy during *P. infestans* infection.

Reviewer #3:

Some of the authors in this manuscript previously found that an effector from the Irish potato famine pathogen *Phytophthora infestans* binds to autophagosome-localized ATG8 protein instead of autophagy cargo receptor Joka2 in order to counteract plant defenses for its infection. In this manuscript Dagdas et al., reported that autophagosomes labeled with GFP-ATG8CL are directed toward the pathogen interface for membrane sources of extra haustorial membrane (EHM).

Although the authors' concept/hypothesis that autophagosomal membranes can be one of the origins and source of the host-microbe interfaces is quite interesting, new mechanistic insights for it are not provided. For example, how are the pathogen-manipulated autophagosomes diverted to the pathogen interface? Additionally, convincing data are still missing to support the authors' claim. If the authors' hypothesis is true, GFP-ATG8CL should be well colocalized with EHM marker REM1.3. GFP-ATG8CL signal is not completely overlapped with REM1.3-RFP signal. It seems to be localized in cytoplasm between EHM and vacuolar membrane. To make a solid paper, immuno-EM analyses using anti-antibodies of autophagosomal membrane marker are necessary.

[Editors’ note: what now follows is the decision letter after the authors submitted for further consideration.]

This manuscript has been improved significantly from previous submission. The work uncovers selective autophagy targeted to haustoria as a defense mechanism, represents one of the first papers to tackle plant cargo receptors, and is a significant contribution to our understanding of how autophagy may contribute immunity. We are happy to offer publication of the manuscript after the authors have addressed the following issues:

1) It is helpful to explain in the Introduction differences of "selective autophagy", "antimicrobial-autophagy", and "xenophagy". This will help readers.

2) Where does ATG8CL localize without infection?

3) The title is a bit misleading by stating "autophagosomes are diverted to the pathogen interface", since it implies initiation of autophagosomes elsewhere then move to haustoria. What the authors show is that autophagosomes are formed around haustoria..

4) Many of the figures show fluorescence of BFP protein fusions using a blue color. Although this is intuitive, it is very difficult to see the blue color against the black background. Since this is a false color, anyhow, I strongly suggest replacing the blue color with something brighter such as yellow or magenta.

5) Figure 4—figure supplement 1 and Figure 4—figure supplement 2 present critical data to support the conclusion that PexRD54 and ATG8CL co-localize in puncta at the perihaustorial membrane. This should be moved to main figures.

6) In the Discussion section the authors allude to phylogenetic analyses that they have performed on the ATG8 family in plants. It would be helpful to explicitly state whether ATG8CL is conserved across flowering plants, or whether it is specific to the Solanaceae family. I gather that it is not broadly conserved. If it is not, do they speculate that other ATG8 members fulfill this role in other plant families?

7) Images in Figure 3 were collected 3-4 days post infection. Based on Figure 3—figure supplement 4, protein levels differ significantly. Please comment whether this affect interpretation of the data.

---

## [Author Response]

[Editors’ note: the author responses to the first round of peer review follow.]

Our decision has been reached after consultation between the reviewers. Based on these discussions and the individual reviews below, we regret to inform you that your work will not be considered further for publication in eLife.There was general agreement that the study is interesting. However, the study falls short of providing enough new mechanistic insights to be suitable for a research advance. Specifically, there are important issues that are not fully substantiated, which raised questions:-Do the events described in the manuscript happen during infection? (That is, what is the biological relevance during natural infection?)-Are autophagosomes actually directed toward the pathogen interface under native conditions? Additional microscopy (e.g. electron microscopy) can help address this important issue.

We are very pleased to submit the manuscript “Host autophagosomes are diverted to the pathogen interface to mediate focal defense responses against the Irish potato famine pathogen” for publication in *eLife* as a Research Advance. We recently showed that the Irish potato famine pathogen *P. infestans* subverts host autophagy related defences by secreting an effector protein (Dagdas et al., eLife, 2016, 5:e10856; Maqbool et al., JBC, 2016, 291:20270). In this manuscript, we provide an exciting cellular biology dimension to our previous model and show that defense-related selective autophagy pathway is diverted to the plant-microbe interface in order to limit the pathogen growth, instead of taking the default vacuolar degradation route. In turn, the pathogen antagonizes the antimicrobial-autophagy launched to destroy itself by deploying an effector protein that disarms the defense-related autophagosomes across the perimicrobial host membrane. Interestingly, the autophagosome biogenesis machinery, marked by the conserved early autophagosome protein ATG9, is also diverted to hostmicrobe interface. Furthermore, we assign a previously unknown antimicrobial function to ATG9 protein, as its overexpression significantly improved plant immunity to pathogen infection.

The extent to which autophagy contributes to immunity and/or serves adapted pathogens is currently under debate. In Dagdas et al. (eLife, 2016), we showed that the effector protein PexRD54 of *P. infestans* outcompetes host autophagy cargo receptor Joka2 in binding the core autophagy adaptor ATG8. This results in suppression of Joka2-mediated immunity. Interestingly, the effector also stimulates the formation of the autophagosomes, which suggests that the pathogen could subvert host autophagy for its own benefit. In this study, we demonstrate that autophagosomes are diverted towards haustoria, specialized hyphal extensions that grow inwards host cells. This reveals a new pathway that is re-routed towards the pathogen interface and provides evidence for extensive remodelling of the host endomembrane system by plant pathogens.

In addition to showing importance of focal autophagic defense responses, our work has wider implications. It sheds light on membrane biogenesis in plant cells that accommodate microbial structures and explains the differences in protein composition between plant plasma membrane and perimicrobial

membranes. This work also points to a possible source for the enigmatic extracellular vesicles that have been reported in host-microbe interfaces of symbiotic and parasitic interactions.

Although the essence of this story is similar to our previous submission that was rejected, this is almost a new manuscript, not a cosmetically re-decorated version of the previous one. We have added several new composite figures and modified the text accordingly. A summary of the new data and figures are as follows:

1) We provide RNAi based genetic evidence demonstrating autophagosome targeting to haustorial interface is dependent on the core autophagy components ATG4 and ATG9. This is now presented in Figure 2.

2) We have performed high resolution imaging of ATG8CL/Joka2/PexRD54 autophagosomes to show that they associate with the perimicrobial host membrane. These data are presented in Figure 1—figure supplement 1, Figure 3—figure supplement 1, and Figure 4—figure supplement 2.

3) We have performed domain-function analyses of Joka2 to show that domains that mediate oligomerization and protein-protein interaction are important for haustorial accumulation of Joka2. Furthermore, we did pathogenicity assays that shows focal accumulation of Joka2 is important for its positive role in antimicrobial immunity. These data are now presented in Figure 3, Figure 3—figure supplement 2, Figure 3—figure supplement 4.

4) We have performed high-resolution imaging and pathogenicity assays that showed perihaustorial niche is a hot spot for autophagosome biogenesis and overexpression of early autophagosome biogenesis marker ATG9 enhances plant immunity. These data are now presented in Figure 5, Figure 5—figure supplement 1.

[Editors' note: the author responses to the re-review follow.]

This manuscript has been improved significantly from previous submission. The work uncovers selective autophagy targeted to haustoria as a defense mechanism, represents one of the first papers to tackle plant cargo receptors, and is a significant contribution to our understanding of how autophagy may contribute immunity. We are happy to offer publication of the manuscript after the authors have addressed the following issues:

We are delighted to hear about the positive evaluation of our manuscript by the reviewers and the editor. We addressed all the raised issues as following:

1) It is helpful to explain in the Introduction differences of "selective autophagy", "antimicrobial-autophagy", and "xenophagy". This will help readers.

We now included new text in the Introduction to clarify different types of autophagy.

2) Where does ATG8CL localize without infection?

We now present images (Figure 1—figure supplement 1A-B) showing punctate and vacuolar localization of ATG8CL in uninfected cells.

3) The title is a bit misleading by stating "autophagosomes are diverted to the pathogen interface", since it implies initiation of autophagosomes elsewhere then move to haustoria. What the authors show is that autophagosomes are formed around huastoria.

We changed the title to “Host autophagy machinery is diverted to the pathogen interface to mediate focal defense responses against the Irish potato famine pathogen”.

4) Many of the figures show fluorescence of BFP protein fusions using a blue color. Although this is intuitive, it is very difficult to see the blue color against the black background. Since this is a false color, anyhow, I strongly suggest replacing the blue color with something brighter such as yellow or magenta.

We thank the editor for this suggestion. We updated all the figures and changed all blue channels to teal and green channels to yellow.

5) Figure 4—figure supplement 1 and Figure 4—figure supplement 2 present critical data to support the conclusion that PexRD54 and ATG8CL co-localize in puncta at the perihaustorial membrane. This should be moved to main figures.

We have revised the figure as suggested. These data are now presented in Figure 4.

6) In the Discussion section the authors allude to phylogenetic analyses that they have performed on the ATG8 family in plants. It would be helpful to explicitly state whether ATG8CL is conserved across flowering plants, or whether it is specific to the Solanaceae family. I gather that it is not broadly conserved. If it is not, do they speculate that other ATG8 members fulfill this role in other plant families?

We included following text in the Discussion “The expansion of plant ATG8 members seems to have occurred earlier in evolution, and each plant family carries a unique set of ATG8 isoforms that have been diversifying over millions of years. Thus, it is likely that in different plant families different ATG8 isoforms are specialized to carry out defense-related tasks.”

7) Images in Figure 3 were collected 3-4 days post infection. Based on Figure 3—figure supplement 4, protein levels differ significantly. Please comment whether this affect interpretation of the data.

We think this is unlikely because we excluded the cells that didn’t look healthy or showed no Joka2 expression. Additionally, we measured the frequency of haustoria that has Joka2 puncta rather than quantifying the abundance of Joka2 puncta/cell. Consistently, our image quantification does not show any significant differences in number of haustoria that accommodate Joka2:BFP puncta in images obtained three (32/35 of haustoria, 91.4%) vs four (14/15 of haustoria 93.3%) days post infection.